# A Comparative Study on Physicochemical, Photocatalytic, and Biological Properties of Silver Nanoparticles Formed Using Extracts of Different Parts of *Cudrania tricuspidata*

**DOI:** 10.3390/nano10071350

**Published:** 2020-07-10

**Authors:** Sun Young Park, Guo Lu, Beomjin Kim, Woo Chang Song, Geuntae Park, Young-Whan Choi

**Affiliations:** 1Bio-IT Fusion Technology Research Institute, Pusan National University, Busan 609-735, Korea; 201210503@pusan.ac.kr (B.K.); dck3202@naver.com (W.C.S.); 2Department of Horticultural Bioscience, Pusan National University, Myrang 627-706, Korea; guolu372888216@pusan.ac.kr; 3Department of Nanomaterials Engineering, Pusan National University, Busan 609-735, Korea; gtpark@pusan.ac.kr

**Keywords:** *Cudrania tricuspidata*, silver nanoparticles, photocatalysis, biological properties

## Abstract

Green-synthesized silver nanoparticles (SNPs) have great potential for biomedical applications, due to their distinctive optical, chemical, and catalytic properties. In this study, we aimed to develop green-synthesized SNPs from extracts of *Cudrania tricuspidata* (CT) roots (CTR), stems (CTS), leaves (CTL), and fruit (CTF) and to evaluate their physicochemical, photocatalytic, and biological properties. CTR, CTS, CTL, and CTF extracts were evaluated and compared for their total phenol and flavonoid content, reducing capacity, and antioxidant activity. The results revealed that CTR, CTS, CTL, and CTF extracts have high phenol and flavonoid content, as well as a powerful antioxidant and reducing capacity. CTR and CTS extracts showed the strongest effects. The results from UV-Vis spectra analysis, dynamic light scattering, high-resolution transmission electron microscopy, energy dispersive spectroscopy, X-ray diffraction, and Fourier-transform infrared spectroscopy showed the successful formation of CT-SNPs with surface morphology, crystallinity, reduction capacity, capsulation, and stabilization. Synthesized CT-SNPs successfully photocatalyzed methylene blue, methyl orange, rhodamine B, and Reactive Black 5 within 20 min. The CTR- and CTS-SNPs showed better antibacterial properties against different pathogenic microbes (*Staphylococcus aureus*, *Bacillus cereus*, *Escherichia coli*, and *Salmonella enteritidis*) than the CTL- and CTF-SNPs. CTS- and CTR-SNPs showed the most effective cytotoxicity and antiapoptosis properties in human hepatocellular carcinoma cells (HepG2 and SK-Hep-1). CT-SNPs also seemed to be more biologically active than the CT extracts. The results of this study provide evidence of the establishment of CT extract SNPs and their physicochemical, photocatalytic, and biological properties.

## 1. Introduction

Silver nanoparticles (SNPs) are noteworthy owing to their extensive applications in cosmetics, biomedical, food, and health industries, fungicidal agents, and water-based systems. This may be due to their unique conductivity, optical catalytic activity, biocompatible nature, and better structural properties than their bulk counterparts [1,2,3]. SNPs have also been shown to possess potential biological properties, such as antibacterial, antioxidant, antitumor, and anti-inflammatory activities [4]. Over the past decade, rapid-green synthesis approaches, and methods for producing SNPs, have been an interesting area in “green” nanotechnologies. Currently, SNP-mediated biogenic sources that are chosen to substitute toxic chemical methods have shown significant potential in reducing polluting reaction by-products. The “green” nanotechnologies are advantageous over physical and chemical methods and exploit economical and ecofriendly methods by using an available and non-toxic medium, such as medicinal plants that leads to the green synthesis of SNPs [5,6,7,8]. The application of medicinal plants in the formation of SNPs has prompted numerous investigations in the field of “green” nanotechnologies. Compared to other biogenic sources, including bacteria and fungi, medicinal plants appear to be more facilitative and faster, which significantly increases the productivity of this green-synthetic approach [9,10]. Most medicinal plant parts, such as roots, stems, leaves, and fruits are used as a medium for the green synthesis of SNPs. Notably, the medicinal plants used were known for their phytochemical characteristics, such as reducing capacity, stabilization, and capping biogenic sources. The major phytochemicals of medicinal plants that affect the reducing capacity, stabilization, and capping of the SNPs are phenolics, flavones, terpenoids, alkaloids, polysaccharides, proteins, and alcoholic compounds [11,12,13].

*Cudrania tricuspidata* (CT) is a member of the *Moraceae* family and is distributed in Korea and East Asia. Different parts of CT including its roots, stems, leaves, and fruit have been widely used for medicinal purposes to treat conditions, such as tumors, liver damage, jaundice, chronic gastritis, rheumatism, as well as external and internal hemorrhage [14,15,16]. Over the past two decades, almost all parts of CT have been pharmaceutically assessed and extensively investigated phytochemically. The pharmacological effects of CT are owing to the presence of a large number of phenolic compounds, including flavonoids, diterpenoids, alkaloids, and terpenoids. In particular, phenolic compounds such as cudratricusxanthones, cudraxanthones, and 1,3,7-trihydroxy-4-(1,1-dimethyl-2-propenyl)-5,6-(2,2-dimethylchromeno)xanthone have antioxidant, antibacterial, anti-inflammatory, and antitumor characteristics. These compounds have already been identified in the roots, stems, leaves, and fruits of CT. It has been demonstrated that different parts (roots, stems, leaves, and fruit) of the CT may exert biological anti-oxidant, anti-bacteria, anti-inflammatory, anti-allergy, anti-obesity, and anti-tumor properties. In particular, the roots and stems of CT have been widely used in Korea and China to treat various diseases like acute arthritis, pulmonary tuberculosis, mumps, and eczema. In Korea and China, its roots and stems are used as traditional Chinese medicine against tumor progression and metastasis in the last few decades [17,18,19,20]. The water-soluble organic moieties of medicinal plants play a significant role in the green synthesis of SNPs using bioactive components with significant redox characteristics. Medicinal plant-based SNPs have been thoroughly analyzed, and it has been experimentally acknowledged that they exhibit medicinal treatments as well as biological effects including anti-oxidative, anti-inflammatory, anti-bacterial, and anti-tumor properties [12,13]. While interest in CT extracts has gradually increased, owing to its various biological properties, little is known about the green synthesis of SNPs from extracts of the various parts of CT. In this study, we first comprehensively applied, compared, and analyzed the composite extracts of CT roots (CTR), stems (CTS), leaves (CTL), and fruit (CTF) on SNPs and performed multiplex assessment of their physicochemical, photocatalytic, and biological properties. To study the powerful antioxidant and reducing capacity of CTR, CTS, CTL, and CTF extracts, we investigated the total phenol and flavonoid content, and reducing power and performed free radical ABTS and DPPH scavenging assays. The physicochemical examination of CTR-, CTS-, CTL-, and CTF-SNPs was performed by UV-Vis spectra analysis, dynamic light scattering (DLS), high-resolution transmission electron microscopy (HR-TEM), energy dispersive spectroscopy (EDS), X-ray diffraction (XRD), and Fourier-transform infrared spectroscopy (FT-IR). UV-Vis spectra analysis was performed, in order to evaluate the photocatalytic degradation of methylene blue (MB), methyl orange (MO), rhodamine B (RB), and Reactive Black 5 (RB5). The biological properties were evaluated to prove antibacterial activity against gram-positive and gram-negative bacteria, as well as antiapoptosis against human hepatocellular carcinoma cells (HepG2 and SK-Hep-1).

## 2. Materials and Methods

### 2.1. Chemicals and Reagents

Silver nitrate (AgNO_3_), MB, MO, RB, RB5, 2,2′-azino-bis(3-ethylbenzothiazoline-6-sulfonic acid) (ABTS), 2,2-Diphenyl-1-picrylhydrazyl (DPPH), Cell Counting Kit-8 (CCK-8), and other chemicals were obtained from Sigma-Aldrich (Saint Louis, MO, USA). The Annexin V FITC Apoptosis Detection Kit was obtained from BD Biosciences (San Jose, CA, USA). All of the other chemicals used were of analytical grade.

### 2.2. Preparation of CT extracts

CT samples were collected from Miryang, Gyungnam Province, Korea, in December 2018. The botanical identification was prepared by Dr. Young Whan Choi (College of Natural Resources and Bioscience, Pusan National University, Korea), and a voucher specimen (No. MT20180011) was deposited at the laboratory of the Natural Products Research Lab., College of Natural Resources and Bioscience, Pusan National University, Korea. Briefly, the roots, stems, leaves, and fruits of CT were powdered using an electric mixer (HMF-3100S; Hanil Electric, Seoul, Korea), after being dried in a hot-air drying machine (JSR, Seoul, Korea) for 24 h at 60 °C. The dried roots, stems, leaves, and fruits (30 g) of CT were then extracted with distilled water at 100 °C for 4 h, filtered, and concentrated using a rotary vacuum evaporator (Buchi Rotavapor R-14; Buchi Labortechnik, Flawil, Switzerland) to acquire 5.467, 2.000, 8.867, and 2.711 g, respectively. The CTR, CTS, CTL, and CTF extracts were dissolved in distilled water, stored as 50 mg/mL stock solutions at 4 °C, and diluted to the chosen concentration using the medium prior to use.

### 2.3. Phenolic and Flavonoid Content of CT Extracts

The total phenolic and flavonoid content of CTR, CTS, CTL, and CTF extracts was determined using the Folin–Ciocalteu and aluminum chloride colorimetric methods, as described previously [21]. The total phenolic and flavonoid content was expressed as gallic acid equivalent (GAE) per g and quercetin equivalent (OE) per g of CTR, CTS, CTL, and CTF extracts.

### 2.4. Free Radical ABTS and DPPH Scavenging Assay of CT Extracts

The free radical ABTS and DPPH scavenging capacity of CTR, CTS, CTL, and CTF extracts was determined according to the method, as described previously [22]. CTR, CTS, CTL, and CTF extracts were mixed with ABTS (7 mM) or DPPH solution (60 µM) in 24-well microplates. The optical density value of the mixture was explored at 734 nm (ABTS) and 510 nm (DPPH), using a FLUOstar^®^ Omega Plate Reader (BMG Labtech, Ortenberg, Germany).

### 2.5. Assay to Determine the Reducing Potential of the CT Extracts

The reducing potential of CTR, CTS, CTL, and CTF extracts was determined, according to the method described by [22]. The reducing potential was expressed as ascorbic acid equivalent (AAE) per g of CTR, CTS, CTL, and CTF extracts. The absorbance was quantified at 700 nm with a spectrophotometer (Evolution™ 300 UV-Vis, Thermo Fisher Scientific, Tulsa, OK, USA).

### 2.6. Synthesis and UV-Vis Spectra Analysis of CT-SNPs

For the reduction of AgNO_3_ to SNPs, 1 mM of aqueous silver nitrate solution was mixed with CT roots (CTR; 4 mg/mL), stems (CTS; 4 mg/mL), leaves (CTL; 4 mg/mL), and fruit extracts (CTF; 4 mg/mL), respectively. The mixture was incubated at 80 °C for 5 min, and the color changed from light yellow to yellowish-brown after 5 min, which confirmed the formation of SNPs. The formation of CTR, CTS, CTL, and CTF-SNPs was detected using an Evolution 300 UV-Vis absorption Spectrophotometer (Thermo Fisher Scientific) from 300 to 800 nm.

### 2.7. Physicochemical Characterization of CT-SNPs

The particle size, zeta-potential, and polydispersity index (PDI) of CTR, CTS, CTL, and CTF-SNPs were determined at 25 °C by DLS technique using Zetasizer Nano ZS90 (Malvern Instruments, Malvern, UK). The CTR, CTS, CTL, and CTF-SNPs were placed in a disposable zeta cell at 25 °C. Distilled water was used as a reference for the dispersing buffer. The results are given as the average particle size, zeta-potential, and PDI acquired from the examination of three different batches, each of them measured three times. The XRD patterns were collected using a X’Pert 3 Powder X-ray Diffractometer (Malvern Panalytical, Malvern, UK) operating at a 30 to 80 s scanning range, 40 kV voltage, and 30 mA current. The FT-IR patterns were collected using a Perkin Elmer Spectrum GX FT-IR Spectrophotometer (Norwalk, CT, USA) with KBr pellets operating at a range of 400 to 4000 cm^−1^. To determine the surface morphology, crystallinity, and chemical composition of CT-SNPs, one drop of the reaction mixture was deposited on the copper grid and then the CT-SNPs were dried completely. The HR-TEM operating at a 200 KV accelerated voltage, as well as the selected area electron diffraction (SAED), fast Fourier transform (FFT), high-angle annular dark field (HAADF), and EDS measurements were conducted on the Thermo Scientific (FEI) Talos F200X G2 TEM instrument (Bremen, Germany). Image J software (TomoJ, SEG3D2, Meshlab) was used to determine the average diameter and size distribution of CT-SNPs.

### 2.8. Photocatalytic Activities of CT-SNPs

The photocatalytic activities of the CTR-, CTS-, CTL-, and CTF-SNPs were evaluated by degrading MB, MO, RB, and RB5 dye by [23]. In brief, CTR-, CTS-, CTL-, and CTF-SNPs were added to MB (0.8 mM), MO (0.05 mM), RB (0.05 mM), and RB5 (0.05 mM) solution and then, ice-cold sodium borohydride (0.06 M) solution was added. The photodegradation monitored using an Evolution™ 300 UV-Vis Spectrophotometer (Thermo Fisher Scientific) in the range of 300 to 800 nm with regular intervals.

### 2.9. Minimum Inhibitory Concentration (MIC) and Minimum Bactericidal Concentration (MBC) Assays

MIC and MBC assays of the CTR-, CTS-, CTL-, and CTF-SNPs were determined using methods according to the clinical and laboratory standards institute guideline [24]. *Staphylococcus aureus* (ATCC 25923, *S. aureus*), *Bacillus subtilis* (ATCC 10783, *B. subtilis*) *Escherichia coli* (ATCC 25922, *E. coli*), and *Salmonella Enteritidis* (ATCC 13076, *S. Enteritidis*) were obtained from the American Type Culture Collection (Rockville, MD, United States). All the bacteria strains were cultured in Mueller Hinton broth (MHB) (MHA, Lab M, Bury, UK) at 35 °C for 24 h on a rotary shaker at 200 rpm. The MIC assay was assessed in 96-well plates using the standard micro-well dilution method. Briefly, serial dilutions (100, 50, 25, 12.5, 6.25, 3.13, 1.56, and 0.78 μg/mL) of CT extracts and CT-SNPs were prepared in 100 μL of MHB into 96-well plates, and then the final inoculum was adjusted to 10^6^ CFU/mL. After 24 h of incubation at 35 °C, the turbidity of the 96-well plates was determined using a FLUOstar Omega Plate Reader (BMG Labtech, Ortenberg, Germany) at 595 nm. The MBC was performed by plating the suspension form each 96-well plate into a MHA plate. The plates were incubated at 37 °C for 24 h, and then the bacterial population was observed.

### 2.10. Cell Culture and Cell Viability

Human hepatocellular carcinoma cell lines (HepG2, SK-Hep-1) were purchased from American Type Culture Collection (ATCC, Manassas, VA, USA). They were cultured with Minimum Essential Medium Eagle (MEM) media (Gibco; Thermo Fisher Scientific, Inc., Waltham, MA, USA) containing 10% FBS (Gibco; Thermo Fisher Scientific, Inc., Waltham, MA, USA), 1% Penicillin/Streptomycin (Gibco; Thermo Fisher Scientific, Inc., Waltham, MA, USA) at 37 °C with 5% CO_2_. Cell viability assay was performed using CCK-8 (Sigma, Saint Louis, MO, USA) solution according to the manufacturer’s instructions. Cells were cultured in 24-well plates at a density of 1 × 10^4^ cells per well for 24 h. Cells were treated with various concentrations (0–100 μg/mL) of CT extracts and CT-SNPs and incubated at 37 °C with 5% CO_2_ for 24 h, 48 h, and 72 h. Afterward, the CCK-8 solution was treated to the medium at 37 °C with 5% CO_2_ for 4 h. The optical density was read at a wavelength of 450 nm with the FLUOstar Omega Plate Reader (BMG Labtech).

### 2.11. Apoptosis Assay

Human hepatocellular carcinoma cell lines (HepG2, SK-Hep-1) seeded in 6-well plates were treated with CT extracts and CT-SNPs (100 μg/mL) then incubated at 37 °C with 5% CO_2_ for 24 h. Afterward, cells were collected by trypsinization, washed with PBS, and suspended in the dilution buffer. Samples were then added in FITC Annexin V/propidium iodide (PI) (BD Sciences, San Jose, CA, USA). The mixture was gently vortexed and incubated for 15 min at room temperature (25 °C) in the dark. Thereafter, early apoptosis, late apoptosis, and necrosis levels were determined based on the intensity that was recorded on a Flow Cytometer (Beckman Coulter FC500, Pasadena, CA, USA).

### 2.12. Statistical Analysis

The data analyses were performed with the Statistical Package for the Social Sciences software, version 17.0 (SPSS Inc, Chicago, IL, USA). Student’s *t*-test and one-way analysis of variance were used to evaluate the differences among groups. A *p*-value of <0.01 or <0.05 was considered to indicate a statistically significant difference.

## 3. Results and Discussion

### 3.1. Optimization of CT-SNPs Using CT Extracts as a Medium

To assess the potential green-synthesized medium of CTR, CTS, CTL, and CTF extracts on the reduction, capsulation, and stabilization of SNPs, the CT extracts were evaluated by five widely used in vitro chemical assays, namely, total phenol content, total flavonoid content, reducing power, ABTS assay, and DPPH assay. The total phenol content of CTR, CTS, CTL, and CTF extracts is presented in Figure 1A. The highest total phenol content was found in CTR extract, while CTF extract had the lowest content. Different from the ranking order of phenol content, the highest flavonoid content was found in the CTR extract, and the lowest in the CTL extract (Figure 1B). The phenol and flavonoid content has a unique chemical power to reduce as well as effectively stabilize and cap SNPs, thereby, preventing their agglomeration [25]. The reducing power as well as the DPPH and ABTS assays were based on the single-electron transfer processes, which could explain their important correlations. The reducing power of CT extracts were in the following order: CTS > CTR > CTL > CTF (Figure 1C). The DPPH values indicated the free radical DPPH scavenging capacity, with the CTR extract showing the strongest capacity and the CTF extract the weakest (Figure 1D).

The order of ABTS values, which indicated the free radical ABTS scavenging capacity, was as follows: CTR > CTS > CTL > CTF (Figure 1E). The total phenol and flavonoid content and free radical scavenging capacity rank orders of CT extracts were quite similar because of similar antioxidant capacities. In addition, the total phenol content of CT extracts easily transferred electrons, and this may be the reason for the redox characteristics of CT extracts with the formation of SNPs. CT extracts showed the strongest overall reducing power activity. It was necessary to determine the potential of the CTR, CTS, CTL, and CTF extracts to reduce the silver nitrate solution that leads to the excellent and controlled formation of SNPs. In this study, the addition of CTR, CTS, CTL, and CTF extracts to an aqueous silver nitrate solution resulted in the formation of CTR-, CTS-, CTL- and CTF-SNPs, respectively. Reduction of Ag^+^ to Ag^0^ facilitated by CTR, CTS, CTL, and CTF extracts was confirmed by the color change of the solution from light yellow to yellowish-brown and by the absorption peak, λmax, observed between 435 and 443 nm when the SNPs were subjected to a surface plasmon resonance band (Figure 1F). The DLS technique was used to determine the hydrodynamic size distribution, zeta potential, and PDI of CTR-, CTS-, CTL-, and CTF-SNPs. The hydrodynamic size distributions were 92.04 ± 2.19 nm, 87.25 ± 3.18 nm, 66.34 ± 4.16 nm, and 75.88 ± 2.66 nm; the zeta potential values were −32.73 ± 1.59 mV, −33.10 ± 0.82 mV, −33.93 ± 0.84 mV, and −28.57 ± 0.62 mV; and the PDI values were 0.25 ± 0.04, 0.22 ± 0.01, 0.25 ± 0.02, and 0.31 ± 0.01 for CTR-, CTS-, CTL-, and CTF-SNPs, respectively (Figure 2A,B). The zeta potential values indicate that CTR-, CTS-, and CTL-SNPs showed higher stability in aqueous solution than CTF-SNPs. PDI is an important parameter for particle distribution, and therefore, CTR-, CTS-, and CTL-SNPs, which had a narrower distribution than CTF-SNPs, showed good distribution in the present study.

### 3.2. HR-TEM Studies of CT-SNPs

The surface morphology of SNPs, in general, is an essential parameter in SNP studies as it exhibits different physicochemical properties, depending on the size and shape. Therefore, the size, shape, and distribution of CTR-, CTS-, CTL-, and CTF-SNPs was explored using HR-TEM. Typical HR-TEM micrographs display nanoparticles with uniform size; the spherical morphology and distribution size of CTR-, CTS-, CTL-, and CTF-SNPs were observed, with 80% of the SNPs presenting a particle size between 20 and 50 nm. Less than 10% of the CT-SNPs were under 20 nm and between 50 and 70 nm (Figure 3A,B). The monodispersed aspect of CTR-, CTS-, CTL-, and CTF-SNPs was attributed to the capping layer of content, which is known to reduce Ag^+^ to Ag^0^ and bind the phenol and flavonoid content to the SNPs. As shown in Figure 3C, the SAED pattern of CTR-, CTS-, CTL-, and CTF-SNPs acquired on a large area displays sets of sharp diffraction spots and concentric rings, which can be indexed with Ag. The FFT pattern of CTR-, CTS-, CTL-, and CTF-SNPs also indicated a face-centered cubic crystal structure, which exhibited bright circular spots [lattice planes of Bragg’s reflection (111), (200), (220), and (311) planes; Figure 3D]. Figure 3E shows selected area high-angle annular dark-field (HAADF) of the CTR-, CTS-, CTL-, and CTF-SNPs and the representative red-particle image for the Ag atoms.

### 3.3. EDS, XRD, and FTIR studies of CT-SNPs

To examine the well-crystallized silver pattern of the CTR-, CTS-, CTL-, and CTF-SNPs, EDS analysis was performed (Figure 4A). The EDS analysis revealed the surface chemical composition of SNPs and the results showed two clear Ag signals pertaining to the elements in CTR-, CTS-, CTL-, and CTF-SNPs. The surface plasmon resonance analysis showed that CTR-, CTS-, CTL-, and CTF-SNPs displayed peaks at 0.24–0.27 keV, and 2.96–2.98 keV, respectively, due to the presence of Ag. The XRD pattern of CTR-, CTS-, CTL-, and CTF-SNPs shows four intense peaks at four 2θ values (30–80°) of 38.12–38.39°, 44.37–44.50°, 64.70–64.89°, and 77.80–77.96°, respectively (Figure 4B). The peaks of the XRD plots seemed to broaden, confirming that the highly purified SNPs are composed of crystalline silver. The peaks at 38.12–38.39°, 44.37–44.50°, 64.70–64.89°, and 77.80–77.96° can be assigned to the typical (111), (200), (220), and (311) planes of the face-centered cubic crystal structure of Ag (JCPDS NO. 65-2871), respectively. No other peaks for any contaminations were detected. This is in line with the FFT pattern and EDS analysis results. The “green” nanotechnologies of SNPs involve medicinal plants, which act as functionalizing ligands under physiological conditions, creating SNPs that are highly appropriate for biological applications. These studies in FT-IR was analyzed to investigate the presence of bioactive components in CTR-, CTS-, CTL-, and CTF-SNPs. The CT extracts also provide possible functional groups, which attach to the surface of the CT-SNPs, resulting in capping and efficient stabilization. In our study, the FT-IR spectra was examined to obtain further evidence about the presence of bioactive components in the CTR-, CTS-, CTL-, and CTF-SNPs such as the capping layer. The FTIR spectra of CTR, CTS, CTL, and CTF extracts and corresponding CTR-, CTS-, CTL-, and CTF-SNPs designated extensive resemblance between the extracts and the SNPs. Furthermore, the prominent peaks of CTR, CTS, CTL, and CTF extracts and their corresponding SNPs are characterized by –OH stretching of alcohols and phenols at 3237.02 cm^−1^, 3238.97 cm^−1^, 3243.28 cm^−1^, and 3259.53 cm^−1^, respectively. –CH stretching of amines was also observed at 2968.89 cm^−1^, 2936.75 cm^−1^, 2896.68 cm^−1^, and 2836.98 cm^−1^, respectively. N–H or C=C bond of phenols or carbonyl groups were also observed at 1635.65 cm^−1^, 1628.42 cm^−1^, 1639.36 cm^−1^, and 1634.68 cm^−1^ (Figure 4C), respectively. These functional groups on CTR-, CTS-, CTL-, and CTF-SNPs show that CTR-, CTS-, CTL-, and CTF-SNPs are stabilized by high –OH stretching or N–H bond possessing bioactive components probably derived from phenols [26]. Based on these results, we suggested that the reduction, capsulation, and stabilization of CTR-, CTS-, CTL-, and CTF-SNPs was owing to functional groups present in the CT extracts, such as phenolics, flavones, terpenoids, alkaloids, polysaccharides, proteins, and alcoholic compounds.

### 3.4. Photocatalytic Studies of CT-SNPs

Organic azo dyes such as MB, MO, RB, and RB5, which are commonly used as an indicator in different methodical fields, were employed [27,28]. The reduction of the MB, MO, RB, and RB5 dyes using sodium borohydride was studied using sodium borohydride in the presence of CTR-, CTS-, CTL-, and CTF-SNPs and monitoring by UV-Vis spectrophotometer. A characteristic absorption peak was observed corresponding to pure MB, MO, RB, and RB5 at 665 nm, 464 nm, 575 nm, and 597 nm, respectively. In the presence of CTR-, CTS-, CTL-, and CTF-SNPs as a catalyst, the deep blue (corresponding to MB), red-yellow (corresponding to MO), pink-red (corresponding to RB), and black-gray (corresponding to RB5) color gradually decreased and finally disappeared. In Figure 5, the UV-Vis spectra revealed that the addition of all the CTR-, CTS-, CTL-, and CTF-SNPs completely decreased in the wavelength of maximum absorbance within 15–20 min. The catalytic reduction of MB was found in following order: CTR-SNPs > CTS-SNPs > CTL-SNPs = CTF-SNPs (Figure 5A), the catalytic reduction of MO was found in following order: CTR-SNPs = CTS-SNPs > CTL-SNPs = CTF-SNPs (Figure 5B), the catalytic reduction of RB was found in following order: CTR-SNPs = CTS-SNPs > CTL-SNPs = CTF-SNPs (Figure 5C), and the catalytic reduction of RB5 was found in following order: CTR-SNPs > CTS-SNPs > CTL-SNPs = CTF-SNPs (Figure 5D). In the presence of CTR, CTL, CTS and CTF extracts only, the intensity of the deep blue (corresponding to MB), red-yellow (corresponding to MO), pink-red (corresponding to RB), and black-gray (corresponding to RB5) color did not alteration (Appendix A). The reaction between the Organic azo dye (acceptor) and sodium borohydride (donor) is related to an electron transfer reaction. The treatment of CT-SNPs having intermediate potential between acceptor and donor makes the electron transfer between Organic azo dye and sodium borohydride very facile. Prior to the degradation, Organic azo dye and sodium borohydride were adsorbed to CT-SNPs. The phytochemical capping CT-SNPs completed this process by attracting Organic azo dye and sodium borohydride towards the CT-SNPs by electrostatic interaction. The good photocatalytic properties of the CT-SNPs might be attributed to their physicochemical characteristics and their functionalization with the CT extract. Previous studies have reported that photocatalytic activity may be contingent on the size, morphology, and crystallographic structure of the SNPs [6,9]. CT-SNP photocatalysts also presented the essential photocatalytic activity to the degradation MB, MO, RB, and RB5 dyes. This can be majorly attributed to the higher surface area of CT-SNPs and synergetic coordination between CT extracts and Ag.

### 3.5. Antibacterial Studies of CT-SNPs

The field of available antibacterial antibiotics can be expanded by “green” nanotechnologies. To compare the total bactericidal and bacteriostatic properties of CT-SNPs against gram-positive (*S. aureus*, *B. cereus*) and gram-negative (*E. coli* and *S. enteritidis*) bacteria with those of the CT extract, we used the microtiter broth dilution method to determine the minimum inhibitory concentration (MIC) and minimum bactericidal concentration (MBC). The MIC was assessed by the visual turbidity of the plates both before, and after, incubation of gram-positive and gram-negative bacteria, and serial dilution was performed to confirm the concentration of CT-SNPs for the MBC. All CT-SNPs under investigation demonstrated outstanding concentration-dependent antibacterial activity against both gram-positive and gram-negative bacteria. The CT-SNPs verified having the highest antibacterial activity against gram-positive and gram-negative bacteria with MIC value of 3.13 to 50 μg/mL and MBC value of 6.25 to 100 μg/mL, a clear suggestion that the MIC value of 3.13 to 12.5 μg/mL and MBC value of 6.25 to 25 μg/mL obtained for CTR-and CTS-SNPs against *S. aureus*, *B. cereus*, *E. coli*, and *S. enteritidis* indicates exceptional antibacterial activity. The CTL-SNPs were active against gram-positive and gram-negative bacteria at a concentration equal to the CTF-SNPs, with an MIC value of 25 μg/mL. Amongst the CT-SNPs investigated in the present study, CTR-SNPs seemed to have most of the great inhibitors with an interesting bactericidal and bacteriostatic properties against gram-positive and gram-negative bacteria (Table 1). On the other hand, no apparent effect of CT extracts (up to 100 μg/mL) was observed for gram-positive and gram-negative bacteria. It is generally accepted that SNPs has a strong biocidal effect against bacteria which inhibits replication of DNA, thus affecting bacterial viability [3,27]. Furthermore, SNPs penetrate the bacterial cell wall and interrupt it. These results indicate that CT-SNPs are broad-spectrum antibacterial candidates which exert the same effect both gram-positive and gram-negative bacteria.

### 3.6. Cytotoxicity Studies of CT-SNPs

Hepatocellular carcinoma is the third leading cause of cancer-associated mortality worldwide. In addition, the survival rate of patients with hepatocellular carcinoma is poor mainly due to resistance, reoccurrence, metastasis, and severe side effects [29,30]. Therefore, extensive efforts are needed to identify novel therapeutic agents and inhibit the hepatocellular carcinoma cells (HepG2, SK-Hep-1) to develop successful long-term treatments. Several studies demonstrated that cytotoxicity of SNPs is not only size type but also the reducing and capping agents use for the synthesis of SNPs. To test our hypothesis, the hepatocellular carcinoma cells (HepG2, SK-Hep-1) were cultured with increasing concentration of CT extracts and CT-SNPs for 24 h, 48 h, and 72 h, followed by the CCK-8 analysis. Individual CTR-, CTS-, CTL-, and CTF-SNPs showed dose dependent cytotoxicity in HepG2 and SK-Hep-1 cell lines, and this was enhanced by time. As expected, CTR-SNPs showed the highest and efficient cytotoxicity, followed by CTF-SNPs then CTL-SNPs, and finally CTF-SNPs in hepatocellular carcinoma cell (HepG2, SK-Hep-1). The IC_50_ values of CTR-, CTS-, CTL-, and CTF-SNPs were calculated to be 37.58 ± 0.23, 42.51 ± 0.37, 83.96 ± 0.32, and 87.29 ± 0.55 μg/mL, respectively. Additionally, the cytotoxic effect in CTR-, CTS-, CTL-, and CTF-SNPs was higher than in the CTR, CTS, CTL, and CTF extracts (Figure 6).

### 3.7. Apoptosis Studies of CT-SNPs

The water-soluble organic moieties of medicinal plants play an important role in the green synthesis of SNPs using phytochemicals with significant antioxidant properties. Medicinal plant-based SNPs have been thoroughly analyzed, and it has been experimentally acknowledged that they exhibit medicinal properties, as well as biological effects, including antitumor activities [31,32]. The effect of CT extract and CT-SNPs on the apoptosis of hepatocellular carcinoma cells (HepG2, SK-Hep-1) was determined by Annexin V-FITC and PI staining assays. Annexin V, which targets the phosphatidylserine on the outer leaflet of the apoptotic cell membrane, can be used as a sensitive probe for the exposure of phosphatidylserine on the apoptotic cell membrane. PI is capable of binding non-specific DNA markers, which is accepted by living cells, and can, thereby, be used to distinguish necrotic cells from apoptotic cells [33,34]. HepG2 and SK-Hep-1 cells were treated with CT extracts and CT-SNPs. After 24 h, cells were stained with Annexin V-FITC and PI for early apoptosis, late apoptosis, and necrosis analysis. As expected, the apoptotic cells increased by CT-SNPs, and CTR-SNPs produced the strongest effects, followed by CTF-SNPs, CTL-SNPs, and CTF-SNPs. CT extract treatment had comparatively little effect on the apoptotic cells. CTR-SNPs induced a higher percentage of early apoptosis (annexin V+/PI-) and late apoptosis (annexin V+/PI+) than other SNPs in HepG2 and SK-Hep-1 cells (Table 2).

## 4. Conclusions

In the present study, SNPs were synthesized using an eco-friendly protocol of extracts from the roots, stems, leaves, and fruit of CT for as reducing, stabilizing, and capping agents for biomedical purposes. This is a one-pot and “green” nanotechnology for quick and facile synthesis of CT-SNPs. The synthesized CTR-, CTS-, CTL-, and CTF-SNPs showed high phenol and flavonoid content, as well as a powerful antioxidant and reducing capacity. The characterization of CTR-, CTS-, CTL-, and CTF-SNPs was conducted using UV-Vis spectra, DLS, HR-TEM, EDS, XRD, and FT-IR. The results showed well dispersed and highly crystalline SNPs. The photocatalytic activity in the CTR-, CTS-, CTL-, and CTF-SNPs during the organic azo dye degradation can be accredited to MB, MO, RB, and RB5, respectively. Our result also demonstrated that CTR-, CTS-, CTL-, and CTF-SNPs have more effective antibacterial properties against gram-positive (*S. aureus, B. cereus*) and gram-negative (*E. coli* and *S. enteritidis*) bacteria and cytotoxicity and apoptosis against hepatocellular carcinoma cells (HepG2, SK-Hep-1) than individual extracts. CTR-SNPs showed significantly more photocatalytic and biological properties than CTS-, CTL-, and CTF-SNPs. SNPs have received increasing attention as anti-bacteria or tumor therapeutics as they can be engineered to target specific bacteria or tumor cells. Recent evidence suggests that anti-bacterial and anti-tumor properties of green synthesized SNPs could be allied with the interaction between SNPs and cell membranes, suppressing the bacteria and tumor cells. According to the results of our study, we suggest that the anti-bacteria and anti-tumor properties of CT-SNPs is an efficient therapeutic target. Further studies are necessary to clarify the underlying potential molecular mechanisms that regulate disease progression involving these anti-bacterial and anti-tumor benefits. Hence, our findings provide strong evidence that CT-SNPs are beneficial owing to their photocatalytic and biological properties and may show promising biomedical applications.

## Figures and Tables

**Figure 1 nanomaterials-10-01350-f001:**
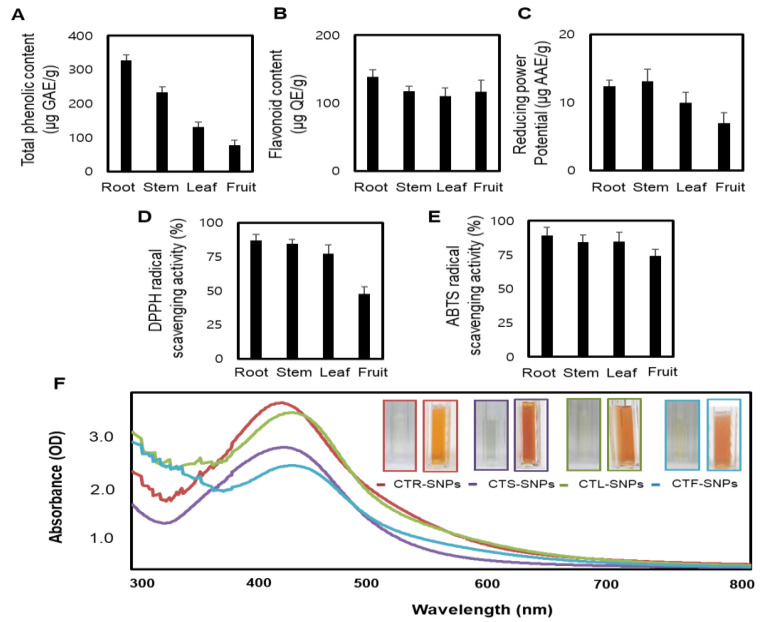
(**A**) Total phenolic content; (**B**) Flavonoid content; (**C**) Reducing potential; (**D**) DPPH radical scavenging activity; (**E**) ABTS radical scavenging activity of CTR, CTS, CTL, and CTF extracts; (**F**) Color change during synthesis (at 80 °C for 5 min) and UV-visible spectrum of CTR, CTS, CTL, and CTF-SNPs.

**Figure 2 nanomaterials-10-01350-f002:**
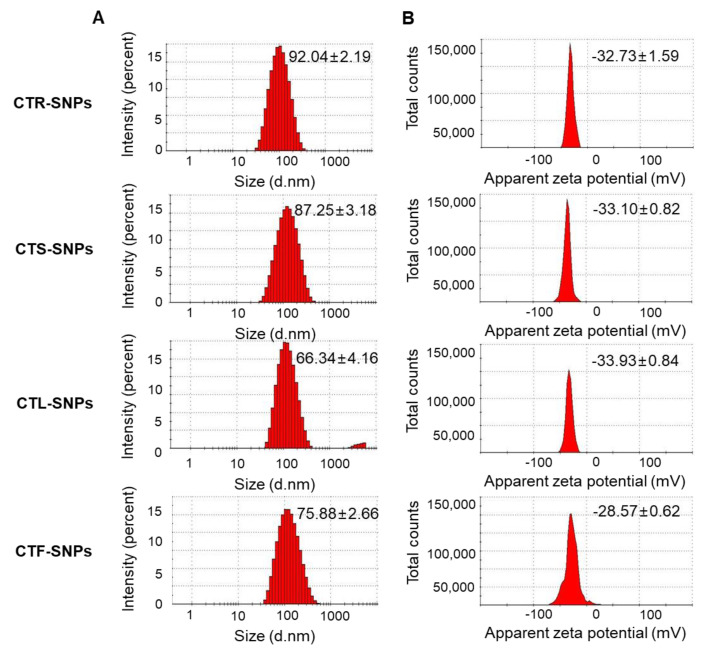
(**A**) DLS showing size distribution and (**B**) zeta potential of CTR, CTS, CTL, and CTF-SNPs.

**Figure 3 nanomaterials-10-01350-f003:**
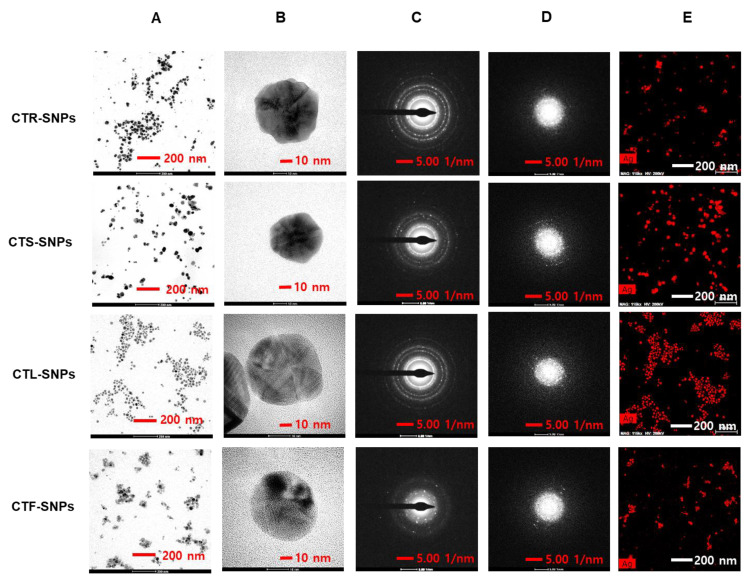
HR-TEM images at (**A**) low magnification, (**B**) high magnification (**C**) SAED pattern, and (**D**) HAADF image, of CTR, CTS, CTL, and CTF-SNPs.

**Figure 4 nanomaterials-10-01350-f004:**
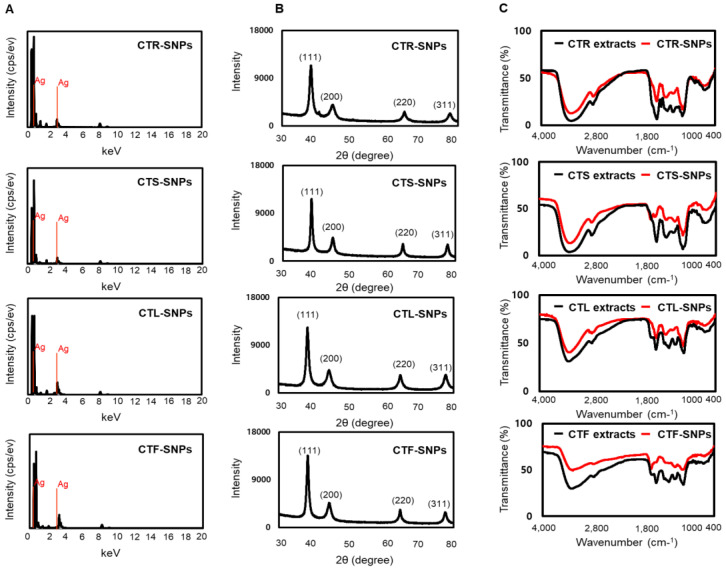
(**A**) EDS analysis, (**B**) XRD pattern, and (**C**) FTIR spectra of CTR, CTS, CTL, and CTF-SNPs.

**Figure 5 nanomaterials-10-01350-f005:**
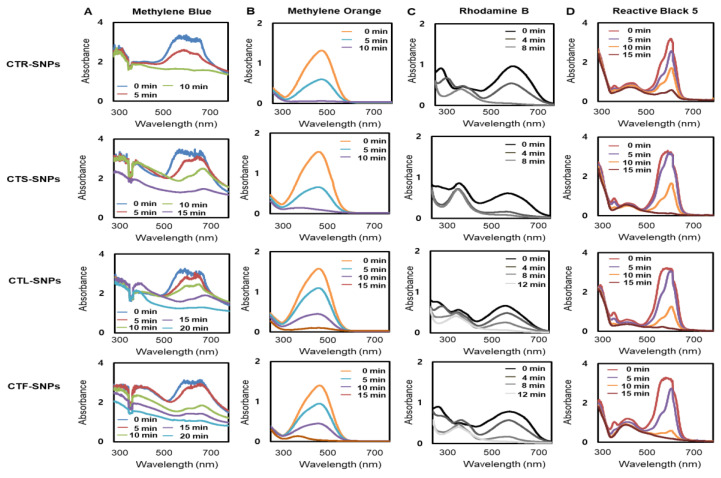
UV-visible spectrum of (**A**) MB, (**B**) MO, (**C**) RB, and (**D**) RB5 on addition of CTR, CTS, CTL, and CTF-SNPs.

**Figure 6 nanomaterials-10-01350-f006:**
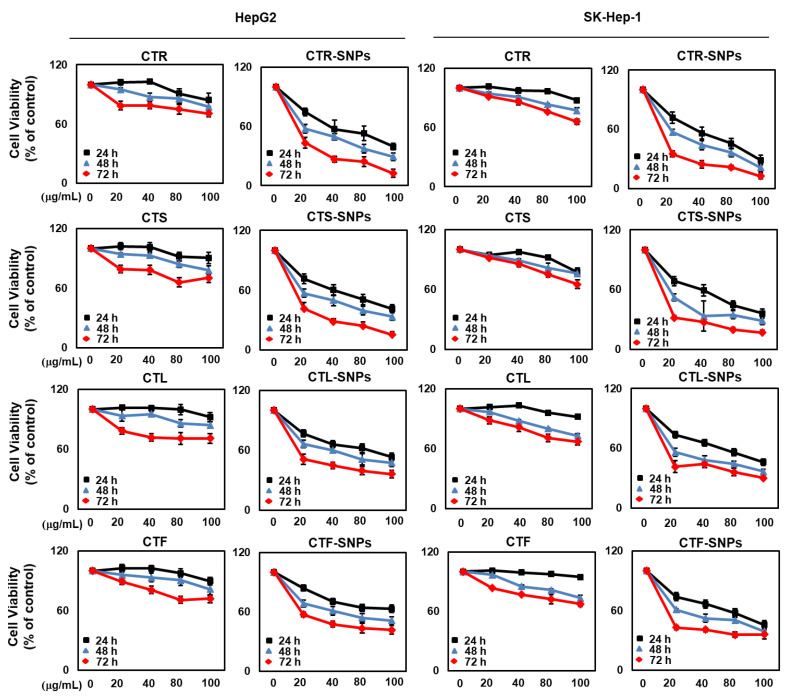
Cytotoxicity of CTR, CTS, CTL, and CTF-SNPs in HepG2 and SK-Hep-1 cells.

**Table 1 nanomaterials-10-01350-t001:** Minimum inhibitory concentration (MIC) and Minimum bactericidal concentration (MBC) of CTR, CTS, CTL and CTF-SNPs.

	Gram Positive Bacteria				Gram Negative Bacteria			
Bacteria	*Staphylococcus aureus*		*Bacillus cereus*		*Escherichina coli*		*Salmonella enteritidis*	
	MIC (μg/mL)	MBC (μg/mL)	MIC (μg/mL)	MBC (μg/mL)	MIC (μg/mL)	MBC (μg/mL)	MIC (μg/mL)	MBC (μg/mL)
CTR	>100	>100	>100	>100	>100	>100	>100	>100
CTR-SNP	6.25	6.25	3.13	12.5	3.13	6.25	6.25	6.25
CTS	>100	>100	>100	>100	>100	>100	>100	>100
CTS-SNP	6.25	12.5	12.5	25	12.5	12.5	6.25	12.5
CTL	>100	>100	>100	>100	>100	>100	>100	>100
CTL-SNP	25	50	25	50	25	50	25	100
CTF	>100	>100	>100	>100	>100	>100	>100	>100
CTF-SNP	25	100	25	100	25	100	25	100

**Table 2 nanomaterials-10-01350-t002:** Apoptosis of CTR, CTS, CTL and CTF-SNPs.

	Early Apoptosis (% of Control)		Late Apoptosis (% of Control)		Necrosis (% of Control)	
Cells	HepG2	SK-Hep-1	HepG2	SK-Hep-1	HepG2	SK-Hep-1
CTR	7.2 ± 1.1	8.3 ± 2.5	3.4 ± 0.9	3.9 ± 0.2	1.1 ± 0.4	1.2 ± 0.6
CTR-SNP	45.6 ± 3.7 **	49.8 ± 6.0 **	38.5 ± 3.0 **	40.5 ± 3.4 **	1.4 ± 0.4	1.7 ± 0.5
CTS	6.4 ± 0.6	7.4 ± 2.3	5.1 ± 1.3	5.9 ± 1.1	2.5 ± 1.3	2.8 ± 0.2
CTS-SNP	40.8 ± 5.3 **	46.2 ± 4.1 **	35.3 ± 4.5 **	40.8 ± 3.8 **	1.2 ± 0.2	1.3 ± 1.0
CTL	4.6 ± 4.3	5.3 ± 1.3	3.2 ± 1.8	3.6 ± 1.0	1.2 ± 0.3	1.3 ± 0.3
CTL-SNP	30.9 ± 3.7 **	35.7 ± 3.3 *	27.4 ± 3.4 **	30.7 ± 2.8 **	1.4 ± 0.2	1.5 ± 0.2
CTF	5.1 ± 1.3	5.4 ± 1.0	3.1 ± 1.5	3.5 ± 0.3	0.9 ± 0.1	1.1 ± 0.2
CTF-SNP	25.2 ± 2.9 **	28.1 ± 3.7 *	15.4 ± 2.6 *	16.6 ± 1.7 *	1.1 ± 0.5	1.4 ± 0.2

* *p* < 0.05 and ** *p* < 0.01 compared to the Control group.

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
