# Peer review of "A Comparative Study on Physicochemical, Photocatalytic, and Biological Properties of Silver Nanoparticles Formed Using Extracts of Different Parts of Cudrania tricuspidata"

_nanomaterials, 2020, doi:10.3390/nano10071350_

Round 1

Reviewer 1 Report

In this manuscript the authors report a study on photocatalytic and biological properties of Silver nanoparticles formed using extract of different part of Cudrania Tricuspidata. The results proposed in this manuscript are interesting and show a good degree of novelty. The paper is generally well written and structured, its contents and results are wellsupported and appropriate for the journal. Nevertheless, the article needs some minor revisions before to be published in Nanomaterials.

Section 1.

In the Introduction sections the authors should refer also to biogenic synthesis inspired to antioxidant and  redox activity of phenolic compounds which represent a valid choice in the synthesis of hybrid organic-inorganic nanostructures.

The authors could improve the references in the introduction section considering the following paper:

Silvestri, B., Armanetti, P., Sanità, G., Vitiello, G., Lamberti, A., Calì, G., Pezzella, A., Luciani, G., Menichetti, L., Luin, S., d’Ischia M. (2019).Silver-nanoparticles as plasmon-resonant enhancers for eumelanin’s photoacoustic signal in a self-structured hybrid nanoprobe. Mater. Sci. Eng. C 102, 788–797.

Silvestri, B., Vitiello, G., Luciani, G., Calcagno, V., Costantini, A., Gallo, M., Parisi, S., Paladino, S. Iacomino, M., D’Errico, G., Caso, M. F., Pezzella, A., d’Ischia M. (2017). Probing the eumelanin-silica interface in chemically engineered bulk hybrid nanoparticles for targeted subcellular antioxidant protection. ACS Appl. Mater. Interfaces 9, 37615–37622. doi: 10.1021/acsami.7b11839

Bhutto, A. A., Kalay, Åž., Sherazi, S. T. H., & Culha, M. (2018). Quantitative structure–activity relationship between antioxidant capacity of phenolic compounds and the plasmonic properties of silver nanoparticles. Talanta189, 174-181.

Du, S., Luo, Y., Liao, Z., Zhang, W., Li, X., Liang, T., Zuo F., Ding, K. (2018). New insights into the formation mechanism of gold nanoparticles using dopamine as a reducing agent. Journal of colloid and interface science523, 27-34.

Section 2.

Pag. 3, Subsection 2.6The authors used a 4mg/ml concentration for all extracts. If possible, clarify the choice of this extracts’ concentration.

Pag.5 Figure 1FIn addition to UV-vis spectra the authors reported photos of the synthesized CT-SNPs colloidal solutions mixtures. I suggest to enlarge these pictures to make the color change more visible. In addition, how long after the beginning of the synthesis both spectra and photos were acquired? The authors refer to visible color changes, of all mixtures, after 5 minutes in Section 2.6. Please add this information in the caption of Figure.

Pag.6 #213Please clarify the ‘positive correlation’ in the sentence: ‘the total phenol content of CT extracts easily transferred electrons, and this may be the reason for the positive correlation with the formation of SNPs’.

Pag.7 Figure 3 The scale bar of all TEM images are not clearly visible. In addition, TEM micrographs show a mean diameter of all CT-NPs between 20 and 50 nm, significantly smaller than the hydrodynamic diameter evaluated by DLS investigation. The authors should better clarify these results.

Pag.7 #255-257 Please replace Au with Ag in the sentence related to EDS analysis.

Pag.8 #266-269 The sentence ‘These studies in FT-IR […] during SNP formation.’ sounds unmeaning, please clarify.

Pag.8 Figure 4CResults about the FT-IR analysis of pure extracts and related CT-SNPs are reported. Changes in the region at about 1500-1700 cm-1are clearly visible comparing the spectrum of pure extract and the corresponding CT-NPs one. These modifications are probably related to the involvement of NH groups in the reduction process, a more detailed discussion related to these results should be added (also citing corresponding references to support data discussion).

Pag.8, Subsection 3.4

Interesting results about the photocatalytic activity of SNPs are reported. Did the authors tested the activity of pure extracts? I suggest, if possible, to compare the activity of CT-NPs with the corresponding free CT in the same amount used in the synthesis of CT-NPs.

Author Response

Thank you for reviewing and providing your comments on my manuscript.

I have revised my manuscript in accordance with the comments from the Reviewer, and my point-by-point responses are listed below.

In this manuscript the authors report a study on photocatalytic and biological properties of Silver nanoparticles formed using extract of different part of Cudrania Tricuspidata. The results proposed in this manuscript are interesting and show a good degree of novelty. The paper is generally well written and structured, its contents and results are wellsupported and appropriate for the journal. Nevertheless, the article needs some minor revisions before to be published in Nanomaterials.

Section 1.

In the Introduction sections the authors should refer also to biogenic synthesis inspired to antioxidant and redox activity of phenolic compounds which represent a valid choice in the synthesis of hybrid organic-inorganic nanostructures.

The authors could improve the references in the introduction section considering the following paper:

Silvestri, B., Armanetti, P., Sanità, G., Vitiello, G., Lamberti, A., Calì, G., Pezzella, A., Luciani, G., Menichetti, L., Luin, S., d’Ischia M. (2019).Silver-nanoparticles as plasmon-resonant enhancers for eumelanin’s photoacoustic signal in a self-structured hybrid nanoprobe. Mater. Sci. Eng. C 102, 788–797.

Silvestri, B., Vitiello, G., Luciani, G., Calcagno, V., Costantini, A., Gallo, M., Parisi, S., Paladino, S. Iacomino, M., D’Errico, G., Caso, M. F., Pezzella, A., d’Ischia M. (2017). Probing the eumelanin-silica interface in chemically engineered bulk hybrid nanoparticles for targeted subcellular antioxidant protection. ACS Appl. Mater. Interfaces 9, 37615–37622. doi: 10.1021/acsami.7b11839

Bhutto, A. A., Kalay, Åž., Sherazi, S. T. H., & Culha, M. (2018). Quantitative structure–activity relationship between antioxidant capacity of phenolic compounds and the plasmonic properties of silver nanoparticles. Talanta, 189, 174-181.

Du, S., Luo, Y., Liao, Z., Zhang, W., Li, X., Liang, T., Zuo F., Ding, K. (2018). New insights into the formation mechanism of gold nanoparticles using dopamine as a reducing agent. Journal of colloid and interface science, 523, 27-34.

=> In accordance with the Reviewer’s comment, we have included the reference. (line 59).

Section 2.

Pag. 3, Subsection 2.6The authors used a 4mg/ml concentration for all extracts. If possible, clarify the choice of this extracts’ concentration.

=> In accordance with the Reviewer’s comment, we have revised this sentence (line 133). “For the reduction of AgNO3 to SNPs, 1 mM of aqueous silver nitrate solution was mixed with CT roots (CTR; 4 mg/mL), stems (CTS; 4 mg/mL), leaves (CTL; 4 mg/mL), and fruit extracts (CTF; 4 mg/mL), respectively.”

Pag.5 Figure 1FIn addition to UV-vis spectra the authors reported photos of the synthesized CT-SNPs colloidal solutions mixtures. I suggest to enlarge these pictures to make the color change more visible. In addition, how long after the beginning of the synthesis both spectra and photos were acquired? The authors refer to visible color changes, of all mixtures, after 5 minutes in Section 2.6. Please add this information in the caption of Figure.

=> In accordance with the Reviewer’s comment, we have revised Figure 1F and caption (line 218). “Figure 1. (A) Total phenolic content, (B) Flavonoid content, (C) Reducing potential (D) DPPH radical scavenging activity (E) ABTS radical scavenging activity of CTR, CTS, CTL, and CTF extracts. (F) Color change during synthesis (at 80 °C for 5 min) and UV-visible spectrum of CTR, CTS, CTL, and CTF-SNPs.”

Pag.6 #213Please clarify the ‘positive correlation’ in the sentence: ‘the total phenol content of CT extracts easily transferred electrons, and this may be the reason for the positive correlation with the formation of SNPs’.

=> In accordance with the Reviewer’s comment, we have revised this sentence (line 224).

“In addition, the total phenol content of CT extracts easily transferred electrons, and this may be the reason for the redox characteristics of CT extracts with the formation of SNPs.”

Pag.7 Figure 3 The scale bar of all TEM images are not clearly visible. In addition, TEM micrographs show a mean diameter of all CT-NPs between 20 and 50 nm, significantly smaller than the hydrodynamic diameter evaluated by DLS investigation. The authors should better clarify these results.

=> In accordance with the Reviewer’s comment, we have revised Figure 3.

Generally, the hydrodynamic size by DLS is higher than that measured from the HR-TEM images as the hydrodynamic size includes the hydration layer in addition to the nature extract on the surface.

Pag.7 #255-257 Please replace Au with Ag in the sentence related to EDS analysis.

=> Thank you for your comment, we have revised typo-error (line 266 and 268).

Pag.8 #266-269 The sentence ‘These studies in FT-IR […] during SNP formation.’ sounds unmeaning, please clarify.

=> In accordance with the Reviewer’s comment, we have revised this sentence (line 277-279).

“These studies in FT-IR were analyzed to investigate the presence of bioactive components in CTR-, CTS-, CTL-, and CTF-SNPs.”

Pag.8 Figure 4CResults about the FT-IR analysis of pure extracts and related CT-SNPs are reported. Changes in the region at about 1500-1700 cm-1are clearly visible comparing the spectrum of pure extract and the corresponding CT-NPs one. These modifications are probably related to the involvement of NH groups in the reduction process, a more detailed discussion related to these results should be added (also citing corresponding references to support data discussion).

=> In accordance with the Reviewer’s comment, we have included in Subsection 2.6 (line 289-291). “These functional groups on CTR-, CTS-, CTL-, and CTF-SNPs show that CTR-, CTS-, CTL-, and CTF-SNPs are stabilized by high –OH stretching or N-H bond possessing bioactive components probably derived from phenols (26). Kahsay, M. H.; RamaDevi, D.; Kumar, Y. P.; Mohan, B. S.; Tadesse, A.; Battu, G.; Basavaiah, K. Synthesis of silver nanoparticles using aqueous extract of Dolichos lablab for reduction of 4-Nitrophenol, antimicrobial and anticancer activities, OpenNano, 2018, 3, 28–37.”

Pag.8, Subsection 3.4

Interesting results about the photocatalytic activity of SNPs are reported. Did the authors tested the activity of pure extracts? I suggest, if possible, to compare the activity of CT-NPs with the corresponding free CT in the same amount used in the synthesis of CT-NPs.

=> In accordance with the Reviewer’s comment, we have included in Supplementary Materials (line 312-315). “In the presence of CTR, CTL, CTS and CTF extracts only, the intensity of the deep blue (corresponding to MB), red-yellow (corresponding to MO), pink-red (corresponding to RB), and black-gray (corresponding to RB5) color did not alteration (Figure S1).”

Reviewer 2 Report

Review on the manuscript nanomaterials-856489 “A Comparative Study on Physicochemical, 2 Photocatalytic, and Biological Properties of Silver 3 Nanoparticles Formed Using Extracts of Different 4 Parts of Cudrania Tricuspidata

Park et al. aimed to develop green-synthesized SNPs from extracts of Cudrania tricuspidata (CT) roots (CTR), stems (CTS), leaves (CTL), and fruit (CTF) and to evaluate their physicochemical, photocatalytic, and biological properties. The results obtained showed the successful formation of CT-SNPs with surface morphology, crystallinity, reduction capacity, capsulation, and stabilization. CT-SNPs successfully photocatalyzed methylene blue, methyl orange, rhodamine B, and Reactive Black 5 within 20 min. The CTR-SNPs and CTS-SNPs showed better antibacterial properties against different pathogenic microbes and were the most effective cytotoxic and antiapoptotic agents in human hepatocellular carcinoma cells (HepG2 and SK-Hep-1). Generally, the article adheres to the journal´s standards.
However, the paper must be improved in following aspects:

  1. Please add in Introduction additional information about the reasons why he prepared

the nanoparticles with the mentioned extracts. What are the advantages of silver nanoparticles compared to the extracts?

  1. Please add in Introduction the literature data about the biological properties of the extracts demonstrated previously. What are the arguments for testing them in this study?
  2. Why did you choose to prepare extracts from all the components of the plant? Is there previous information about the differences regarding the biological effects?
  3. The authors should specify more clearly what the meanings in biology are for the catalytic studies.
  4. The “Discussion” chapter should present comparative the obtained results in this study with results of other studies which used Cudrania Tricuspidata extracts and silver nanoparticles prepared with these extracts.
  5. The authors should specify the mechanisms involved in the antibacterial and antitumor effects. The study focuses more on the preparation and characterization of the compounds and on the finding of some of their effects without any explanations regarding the mechanisms involved in these effects.

Author Response

Park et al. aimed to develop green-synthesized SNPs from extracts of Cudrania tricuspidata (CT) roots (CTR), stems (CTS), leaves (CTL), and fruit (CTF) and to evaluate their physicochemical, photocatalytic, and biological properties. The results obtained showed the successful formation of CT-SNPs with surface morphology, crystallinity, reduction capacity, capsulation, and stabilization. CT-SNPs successfully photocatalyzed methylene blue, methyl orange, rhodamine B, and Reactive Black 5 within 20 min. The CTR-SNPs and CTS-SNPs showed better antibacterial properties against different pathogenic microbes and were the most effective cytotoxic and antiapoptotic agents in human hepatocellular carcinoma cells (HepG2 and SK-Hep-1). Generally, the article adheres to the journal´s standards.

However, the paper must be improved in following aspects:

1.Please add in Introduction additional information about the reasons why he prepared the nanoparticles with the mentioned extracts.

=> In accordance with the Reviewer’s comment, we have included in Introduction (line 75-80). “The water-soluble organic moieties of medicinal plants play a significant role in the green synthesis of SNPs using bioactive components with significant redox characteristics. Medicinal plant-based SNPs have been thoroughly analyzed, and it has been experimentally acknowledged that they exhibit medicinal treatments as well as biological effects including anti-oxidative, anti-inflammatory, anti-bacterial, and anti-tumor properties (12, 13).” While interest in CT extracts has gradually increased owing to its various biological properties, little is known about the green synthesis of SNPs from extracts of the various parts of CT. In this study, we first comprehensively applied, compared, and analyzed the composite extracts of CT roots (CTR), stems (CTS), leaves (CTL), and fruit (CTF) on SNPs and performed multiplex assessment of their physicochemical, photocatalytic, and biological properties.

What are the advantages of silver nanoparticles compared to the extracts?

=> Our results demonstrated that CTR-, CTS-, CTL-, and CTF-SNPs had more effective effects than CTR, CTL, CTS and CTF extracts on organic azo dye degradation, anti-gram-positive (S. aureus, B. cereus) and gram-negative (E. coli and S. enteritidis) bacteria effects, and cytotoxicity and apoptosis against hepatocellular carcinoma cells (HepG2, SK-Hep-1).

2.Please add in Introduction the literature data about the biological properties of the extracts demonstrated previously.

=> In accordance with the Reviewer’s comment, we have included in Introduction (line 70-75). “It has been demonstrated that different parts (roots, stems, leaves, and fruit) of the CT may exert biological anti-oxidant, anti-bacteria, anti-inflammatory, anti-allergy, anti-obesity, and anti-tumor properties. In particular, the roots and stems of CT have been widely used in Korea and China to treat various diseases like acute arthritis, pulmonary tuberculosis, mumps, and eczema. In Korea and China, its roots and stems are used as traditional Chinese medicine against tumor progression and metastasis in the last few decades (17-20).”

What are the arguments for testing them in this study?

=> In this study, we first comprehensively applied, compared, and analyzed the composite extracts of CT roots (CTR), stems (CTS), leaves (CTL), and fruit (CTF) on SNPs and performed multiplex assessment of their physicochemical, photocatalytic, and biological properties.

3.Why did you choose to prepare extracts from all the components of the plant? Is there previous information about the differences regarding the biological effects?

=> Cudrania tricuspidata (CT) is a member of the Moraceae family and is distributed in Korea and East Asia. Different parts of CT including its roots, stems, leaves, and fruit have been widely used for medicinal purposes to treat conditions such as tumors, liver damage, jaundice, chronic gastritis, rheumatism, as well as external and internal hemorrhage (14-16). Over the past two decades, almost all parts of CT have been pharmaceutically assessed and extensively investigated phytochemically. The pharmacological effects of CT are owing to the presence of a large number of phenolic compounds, including flavonoids, diterpenoids, alkaloids, and terpenoids. In particular, phenolic compounds such as cudratricusxanthones, cudraxanthones, and 1,3,7-trihydroxy-4-(1,1-dimethyl-2-propenyl)-5,6-(2,2-dimethylchromeno)xanthone have antioxidant, antibacterial, anti-inflammatory, and antitumor characteristics. These compounds have already been identified in the roots, stems, leaves, and fruit of CT. It has been demonstrated that different parts (roots, stems, leaves, and fruit) of the CT may exert biological anti-oxidant, anti-bacteria, anti-inflammatory, anti-allergy, anti-obesity, and anti-tumor properties. In particular, the roots and stems of CT have been widely used in Korea and China to treat various diseases like acute arthritis, pulmonary tuberculosis, mumps, and eczema. In Korea and China, its roots and stems are used as traditional Chinese medicine against tumor progression and metastasis in the last few decades (17-20). The water-soluble organic moieties of medicinal plants play a significant role in the green synthesis of SNPs using bioactive components with significant redox characteristics. Medicinal plant-based SNPs have been thoroughly analyzed, and it has been experimentally acknowledged that they exhibit medicinal treatments as well as biological effects including anti-oxidative, anti-inflammatory, anti-bacterial, and anti-tumor properties (12, 13).

=> So, the results of the present study demonstrated that the preparation of SNPs from CTR, CTS, CTF, and CTL extracts is more beneficial than using conventional methods as they contain high phenolic compounds, which are accountable for the reduction, encapsulation, and stabilization of SNPs.

4.The authors should specify more clearly what the meanings in biology are for the catalytic studies.

=> In accordance with the Reviewer’s comment, we have included in Subsection 3.4 (line 322-326). “Previous studies have reported that photocatalytic activity may be contingent on the size, morphology, and crystallographic structure of the SNPs (6, 9). CT-SNP photocatalysts also presented the essential photocatalytic activity to the degradation MB, MO, RB, and RB5 dyes. This can be majorly attributed to the higher surface area of CT-SNPs and synergetic coordination between CT extracts and Ag.”

5.The “Discussion” chapter should present comparative the obtained results in this study with results of other studies which used Cudrania Tricuspidata extracts and silver nanoparticles prepared with these extracts.

=> To the best of our knowledge, this is the first detailed study on the assessment of the synthesis and physicochemical characteristics of CTR-SNPs, CTS-SNPs, CTL-SNPs, and CTF-GNPs.

6.The authors should specify the mechanisms involved in the antibacterial and antitumor effects. The study focuses more on the preparation and characterization of the compounds and on the finding of some of their effects without any explanations regarding the mechanisms involved in these effects.

=> our findings provide strong evidence that CT-SNPs has anti-bacterial and anti-inflammatory activity. however, the mechanisms of action of CT-SNPs still require further studies for better understanding.

Round 2

Reviewer 2 Report

Comments and suggestions: The authors have satisfactory answered to all questions mentioned.

Author Response

Thank you for reviewing and providing your comments on my manuscript.